# Causal-BALD: Deep Bayesian Active Learning of Outcomes to Infer Treatment-Effects from Observational Data

**Andrew Jesson**[*]
OATML
University of Oxford

**Panagiotis Tigas**[*]
OATML
University of Oxford

**Joost van Amersfoort**
OATML
University of Oxford

**Andreas Kirsch**
OATML
University of Oxford

**Uri Shalit**
Machine Learning and Causal Inference in Healthcare Lab
Technion

**Yarin Gal**
OATML
University of Oxford

## Abstract

Estimating personalized treatment effects from high-dimensional observational data is essential in situations where experimental designs are infeasible, unethical, or expensive. Existing approaches rely on fitting deep models on outcomes observed for treated and control populations. However, when measuring individual outcomes is costly, as is the case of a tumor biopsy, a sample-efficient strategy for acquiring each result is required. Deep Bayesian active learning provides a framework for efficient data acquisition by selecting points with high uncertainty. However, existing methods bias training data acquisition towards regions of non-overlapping support between the treated and control populations. These are not sample-efficient because the treatment effect is not identifiable in such regions. We introduce causal, Bayesian acquisition functions grounded in information theory that bias data acquisition towards regions with overlapping support to maximize sample efficiency for learning personalized treatment effects. We demonstrate the performance of the proposed acquisition strategies on synthetic and semi-synthetic datasets IHDP and CMNIST and their extensions, which aim to simulate common dataset biases and pathologies.

## 1 Introduction

How will a patient's health be affected by taking a medication [37]? How will a user's question be answered by a search recommendation [34]? We can gain insight into these questions by learning about personalized treatment effects. Estimating personalized treatment effects from observational data is essential when experimental designs are infeasible, unethical, or expensive. Observational data represent a population of individuals described by a set of pre-treatment covariates (age, blood pressure, socioeconomic status), an assigned treatment (medication, no medication), and a post-treatment outcome (severity of migraines). An ideal personalized treatment effect is the difference between the post-treatment outcome if the individual receives treatment and the post-treatment outcome if they do not receive treatment. However, it is impossible to observe both outcomes for an individual; therefore, the difference is estimated between populations instead. In the setting of binary treatments, data belong to either the *treatment group* (individuals that received the treatment) or the *control group* (individuals who did not). The personalized treatment effect is the expected difference

---

[*]Equal contribution. Correspondence to {`andrew.jesson, panagiotis.tigas`}@cs.ox.ac.uk

35th Conference on Neural Information Processing Systems (NeurIPS 2021).

in outcomes between treated and controlled individuals who share the same (or similar) measured covariates; as an illustration, see the difference between the solid lines in Fig. 1 (middle pane).

The use of pre-treatment covariates assembled from high-dimensional, heterogeneous measurements such as medical images and electronic health records is increasing [44]. Deep learning methods have been shown capable of learning personalized treatment effects from such data [42, 43, 21]. However, a problem in deep learning is data efficiency. While modern methods are capable of impressive performance, they need a significant amount of labeled data. Acquiring labeled data can be expensive, requiring specialist knowledge or an invasive procedure to determine the outcome. Therefore, it is desirable to minimize the amount of labeled data needed to obtain a well-performing model. Active learning provides a principled framework to address this concern [8]. In active learning for treatment effects [10, 45, 39] a model is trained on available labeled data consisting of covariates, assigned treatments, and acquired outcomes. The model predictions decide the most informative examples from data comprised of only covariates and treatment indicators. Outcomes are acquired, e.g., by performing a biopsy for the selected patients, and the model is retrained and evaluated. This process repeats until either a satisfactory performance level is achieved or the labeling budget is exhausted.

At first sight, this might seem simple; however, active learning induces biases that result in divergence between the distribution of the acquired training data and the distribution of the pool set data [13]. In the context of learning causal effects, such bias can have both positive and negative consequences. For example, while random acquisition active learning results in an unbiased sample of the training data, we demonstrate how it can lead to over-allocation of resources to the mode of the data at the expense of learning about underrepresented data. Conversely, while biasing acquisitions toward lower density regions of the pool data can be desirable, it can also lead to outcome acquisition for data with unidentifiable treatment effects, which leads to uninformed, potentially harmful, personalized recommendations.

To see how training data bias can benefit treatment effect estimation, consider one difference between experimental and observational data: the treatment assignment mechanism is unavailable for observational data. In observational data, variables that affect treatment assignment (an untestable condition) may be unobserved. Moreover, the relative proportion of treated to controlled individuals varies across different sub-populations of the data. Fig. 1 illustrates the latter point, where there are relatively equal proportions of treated and controlled examples for data in region 3. However, the proportions become less balanced as we move to either the left or the right. In extreme cases, say if a group described by some covariate values were systematically excluded from treatment, the treatment effect for that group *cannot be known* [38]. Fig. 1 illustrates this in region 1, where only controlled examples reside, and in region 5, where only treated cases occur. In the language of causal inference, the necessity of seeing both treated and untreated examples for each sub-population corresponds to satisfying the overlap (or positivity) assumption (see 2.3). The data available in the pool set limits overlap when treatments cannot be assigned. In this setting, regions 2 and 4 of Fig. 1 are very interesting because while either the treated or control group are underrepresented, there may still

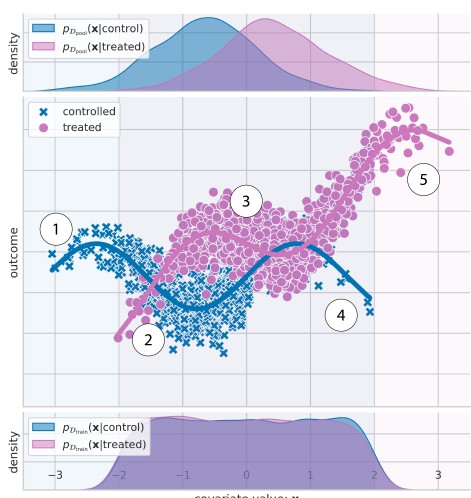

Figure 1: Observational data. Top: data density of treatment (right) and control (left) groups. Middle: observed outcome response for treatment (circles) and control (x's) groups. Bottom: data density for active learned training set after a number of acquisition steps.

be sufficient coverage to estimate treatment effects. D'Amour and Franks [9] have described such regions as having weak overlap. Training data bias towards such regions can benefit treatment effect estimation for underrepresented data by acquiring low-frequency data with sufficient overlap.

We hypothesize that the efficient acquisition of unlabeled data for treatment effect estimation focuses on only exploring regions with sufficient overlap, and uncertainty should be high for areas with non-overlapping support. The bottom pane of Fig. 1 imagines what a resulting training set distribution

could look like at an intermediate active learning step. It is not trivial to design such acquisition functions: naively applying active learning acquisition functions results in suboptimal and sample inefficient acquisitions of training examples, as we show below. To this end, we develop epistemic uncertainty-aware methods for active learning of personalized treatment effects from high dimensional observational data. In contrast to previous work that uses only information gain as the acquisition objective, we propose $\rho$BALD and $\mu\rho$BALD as "Causal BALD" objectives because they consider both the information gain and overlap between treated and control groups. We demonstrate the performance of the proposed acquisition strategies using synthetic and semi-synthetic datasets.

## 2 Background

### 2.1 Estimation of Personalized Treatment-Effects

Personalized treatment-effect estimation seeks to know the effect of a treatment $T \in \mathcal{T}$ on the outcome $Y \in \mathcal{Y}$ for individuals described by covariates $\mathbf{X} \in \mathcal{X}$. In this work, we consider the random variable (r.v.) T to be binary ($\mathcal{T} = \{0, 1\}$), the r.v. Y to be part of a bounded set $\mathcal{Y}$, and $\mathbf{X}$ to be a multi-variate r.v. of dimension $d$ ($\mathcal{X} = \mathbb{R}^d$). Under the Neyman-Rubin causal model [33, 41], the individual treatment effect (ITE) for a person $u$ is defined as the difference in potential outcomes $Y^1(u) - Y^0(u)$, where the r.v. $Y^1$ represents the potential outcome were they *treated*, and the r.v. $Y^0$ represents the potential outcome were they *controlled* (not treated). Realizations of the random variables $\mathbf{X}$, T, Y, $Y^0$, and $Y^1$ are denoted by $\mathbf{x}$, t, y, $y^0$, and $y^1$, respectively.

The ITE is a fundamentally unidentifiable quantity, so instead we look at the expected difference in potential outcomes for individuals described by $\mathbf{X}$, or the Conditional Average Treatment Effect (CATE): $\tau(\mathbf{x}) \equiv \mathbb{E}[Y^1 - Y^0 \mid \mathbf{X} = \mathbf{x}]$ [1]. The CATE is identifiable from an observational dataset $\mathcal{D} = \{(\mathbf{x}_i, t_i, y_i)\}_{i=1}^n$ of samples $(\mathbf{x}_i, t_i, y_i)$ from the joint empirical distribution $P_\mathcal{D}(\mathbf{X}, T, Y^0, Y^1)$, under the following three assumptions [41]:

**Assumption 2.1.** *(Consistency)* $y = ty^t + (1 - t)y^{1-t}$, *i.e. an individual's observed outcome* y *given assigned treatment* t *is identical to their potential outcome* $y^t$.

**Assumption 2.2.** *(Unconfoundedness)* $(Y^0, Y^1) \perp\!\!\!\perp T \mid \mathbf{X}$.

**Assumption 2.3.** *(Overlap)* $0 < \pi_t(\mathbf{x}) < 1 : \forall t \in \mathcal{T}$,

where $\pi_t(\mathbf{x}) \equiv P(T = t \mid \mathbf{X} = \mathbf{x})$ is the **propensity for treatment** for individuals described by covariates $\mathbf{X} = \mathbf{x}$. When these assumptions are satisfied, $\hat{\tau}(\mathbf{x}) \equiv \mathbb{E}[Y \mid T = 1, \mathbf{X} = \mathbf{x}] - \mathbb{E}[Y \mid T = 0, \mathbf{X} = \mathbf{x}]$ is an unbiased estimator of $\tau(\mathbf{x})$ and is identifiable from observational data.

A variety of parametric [40, 46, 42] and non-parametric estimators [17, 49, 3, 14] have been proposed for CATE. Here, we focus on parametric estimators for compactness. Parametric CATE estimators assume that outcomes y are generated according to a likelihood $p_{\boldsymbol{\omega}}(y \mid \mathbf{x}, t)$, given measured covariates $\mathbf{x}$, observed treatment t, and model parameters $\boldsymbol{\omega}$. For continuous outcomes, a Gaussian likelihood can be used: $\mathcal{N}(y \mid \hat{\mu}_{\boldsymbol{\omega}}(\mathbf{x}, t), \hat{\sigma}_{\boldsymbol{\omega}}(\mathbf{x}, t))$. For discrete outcomes, a Bernoulli likelihood can be used: $\text{Bern}(y \mid \hat{\mu}_{\boldsymbol{\omega}}(\mathbf{x}, t))$. In both cases, $\hat{\mu}_{\boldsymbol{\omega}}(\mathbf{x}, t)$ is a parametric estimator of $\mathbb{E}[Y \mid T = t, \mathbf{X} = \mathbf{x}]$, which leads to: $\hat{\tau}_{\boldsymbol{\omega}}(\mathbf{x}) \equiv \hat{\mu}_{\boldsymbol{\omega}}(\mathbf{x}, 1) - \hat{\mu}_{\boldsymbol{\omega}}(\mathbf{x}, 0)$, a parametric CATE estimator.

Jesson et al. [20] have shown that Bayesian inference over the model parameters $\boldsymbol{\omega}$, treated as stochastic instances of the random variable $\boldsymbol{\Omega} \in \mathcal{W}$, yields a model capable of quantifying when assumption 2.3 (overlap) does not hold. Moreover, they show that such models can quantify when there is insufficient knowledge about the treatment effect $\tau(\mathbf{x})$ because the observed value $\mathbf{x}$ lies far from the support of $P_\mathcal{D}(\mathbf{X}, T, Y^0, Y^1)$. Such methods seek to enable sampling from the posterior distribution of the model parameters given the data, $p(\boldsymbol{\Omega} \mid \mathcal{D})$. Each sample, $\boldsymbol{\omega} \sim p(\boldsymbol{\Omega} \mid \mathcal{D})$ induces a unique CATE function $\hat{\tau}_{\boldsymbol{\omega}}(\mathbf{x})$. Jesson et al. [20] propose $\text{Var}_{\boldsymbol{\omega} \sim p(\boldsymbol{\Omega} \mid \mathcal{D})}(\hat{\mu}_{\boldsymbol{\omega}}(\mathbf{x}, 1) - \hat{\mu}_{\boldsymbol{\omega}}(\mathbf{x}, 0))$ as a measure of epistemic uncertainty (i.e., how much the functions "disagree" with one another at a given value $\mathbf{x}$) [23] for the CATE estimator.

### 2.2 Active Learning

Formally, an active learning setup consists of an unlabeled dataset $\mathcal{D}_{\text{pool}} = \{\mathbf{x}_i\}_{i=1}^{n_{\text{pool}}}$, a labeled training set $\mathcal{D}_{\text{train}} = \{\mathbf{x}_i, y_i\}_{i=1}^{n_{\text{train}}}$, and a predictive model with likelihood $p_{\boldsymbol{\omega}}(y \mid \mathbf{x})$ parameterized by $\boldsymbol{\omega} \sim p(\boldsymbol{\Omega} \mid \mathcal{D}_{\text{train}})$. The setup also assumes that an oracle exists to provide outcomes y for any

data point in $\mathcal{D}_{\text{pool}}$. After model training, a batch of data $\{\mathbf{x}_i^*\}_{i=1}^b$ is selected from $\mathcal{D}_{\text{pool}}$ using an acquisition function $a$ according to the informativeness of the batch.

By including the treatment, we depart from the standard active learning setting. For active learning of treatment effects, we define $\mathcal{D}_{\text{pool}} = \{\mathbf{x}_i, t_i\}_{i=1}^{n_{\text{pool}}}$, a labeled training set $\mathcal{D}_{\text{train}} = \{\mathbf{x}_i, t_i, y_i\}_{i=1}^{n_{\text{train}}}$, and a predictive model with likelihood $p_{\boldsymbol{\omega}}(y \mid \mathbf{x}, t)$ parameterized by $\boldsymbol{\omega} \sim p(\boldsymbol{\Omega} \mid \mathcal{D}_{\text{train}})$. The acquisition function takes as input $\mathcal{D}_{\text{pool}}$ and returns a batch of data $\{\mathbf{x}_i, t_i\}_{i=1}^b$ which are labelled using an oracle and added to $\mathcal{D}_{\text{train}}$. We are specifically examining the case when there is access to only the treatments observed in the pool data $\{t_i\}_{i=1}^{n_{\text{pool}}}$: i.e., scenarios where treatment assignment is not possible.

An intuitive way to define informativeness is using the estimated uncertainty of our model. In general, we can distinguish two sources of uncertainty: epistemic and aleatoric uncertainty [11, 23]. Epistemic (or model) uncertainty, arises from ignorance about the model parameters. For example, this is caused by the model not seeing similar data points during training, so it is unclear what the correct label would be. We focus on using epistemic uncertainty to identify the most informative points for label acquisition.

**Bayesian Active Learning by Disagreement (BALD)** [18] defines an acquisition function based on epistemic uncertainty. Specifically, it uses the mutual information (MI) between the unknown output and model parameters as a measure of disagreement:

$$\mathrm{I}(Y; \boldsymbol{\Omega} \mid \mathbf{x}, \mathcal{D}_{\text{train}}) = \mathrm{H}(Y \mid \mathbf{x}, \mathcal{D}_{\text{train}}) - \mathbb{E}_{\boldsymbol{\omega} \sim p(\boldsymbol{\Omega} \mid \mathcal{D}_{\text{train}})} \left[ \mathrm{H}(Y \mid \mathbf{x}, \boldsymbol{\omega}) \right], \qquad (1)$$

where H is the entropy function; a straightforward estimand for discrete outcomes with Bernoulli or Categorical likelihoods.

The general acquisition function based on BALD for acquiring a batch of data points given the pool dataset and the model parameters is given by the joint mutual information between the set $\{Y_i\}$ and the model parameters [26]:

$$a_{\text{BALD}}(\mathcal{D}_{\text{pool}}, p(\boldsymbol{\Omega} \mid \mathcal{D}_{\text{train}})) = \underset{\{\mathbf{x}_i\}_{i=1}^b \subseteq \mathcal{D}_{\text{pool}}}{\arg\max} \; \mathrm{I}(\{Y_i\}; \boldsymbol{\Omega} \mid \{\mathbf{x}_i\}, \mathcal{D}_{\text{train}}). \qquad (2)$$

This batch acquisition function can be upper-bounded by scoring each point in $\mathcal{D}_{\text{pool}}$ independently and taking the top $b$; however, this bound ignores correlations between the samples. In fact, for datasets with significant repetition, this approach can perform worse than random acquisition, and computing the joint mutual information (introduced as *BatchBALD*) rectifies the issue [26].

Estimating the joint mutual information is computationally expensive, as evaluating the joint entropy over all possible outcomes (for classification) or a covariance matrix over all inputs (for regression) is required. An alternative approach is to use softmax-BALD, which involves importance weighted sampling across $\mathcal{D}_{\text{pool}}$ with the individual importance weights given by BALD [25]. We use softmax-BALD for batch acquisition because it is computationally more efficient and performs competitively with BatchBALD. We discuss how BALD maps onto epistemic uncertainty quantification in CATE and the arising complications stemming from the question of overlap in Section 3.

## 3 Methods

In this section: we introduce several acquisition functions, we then analyze how they bias the acquisition of training data, and we show the resulting CATE functions learned from such training data. We are interested in acquisition functions conditioned on realizations of both $\mathbf{x}$ and $t$:

$$a(\mathcal{D}_{\text{pool}}, p(\boldsymbol{\Omega} \mid \mathcal{D}_{\text{train}})) = \underset{\{\mathbf{x}_i, t_i\}_{i=1}^b \subseteq \mathcal{D}_{\text{pool}}}{\arg\max} \; \mathrm{I}(\bullet \mid \{\mathbf{x}_i, t_i\}, \mathcal{D}_{\text{train}}), \qquad (3)$$

where $\mathrm{I}(\bullet \mid \mathbf{x}, t, \mathcal{D}_{\text{train}})$ is a measure of disagreement between parametric function predictions given $\mathbf{x}$ and $t$ over samples $\boldsymbol{\omega} \sim p(\boldsymbol{\Omega} \mid \mathcal{D})$. We make assumptions 2.1 and 2.2 (consistency, and unconfoundedness). We relax assumption 2.3 (overlap) by allowing for its violation over subsets of the support of $\mathcal{D}_{\text{pool}}$. We present all theorems, proofs, and detailed assumptions in Appendix A.

### 3.1 How do naive acquisition functions bias the training data?

To motivate Causal–BALD, we first look at a set of naive acquisition functions. A random acquisition function selects data points uniformly at random from $\mathcal{D}_{\text{pool}}$ and adds them to $\mathcal{D}_{\text{train}}$. In Fig. 2a we

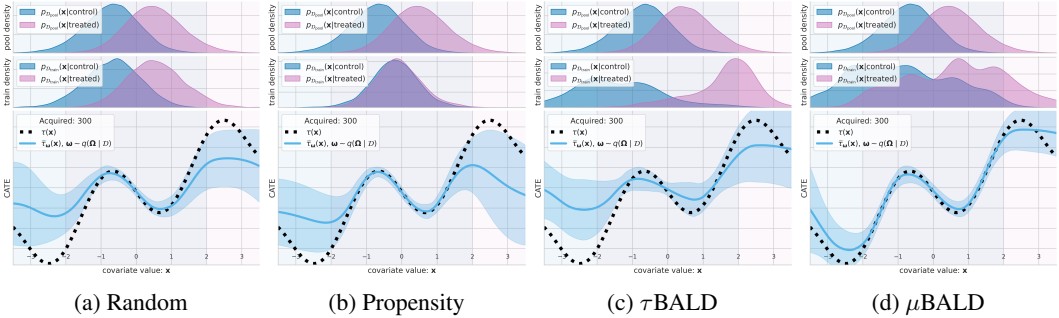

Figure 2: Naive acquisition functions: How the training set is biased and how this effects the CATE function with a fixed budget of 300 acquired points.

have acquired 300 such examples from a synthetic dataset and trained a deep-kernel Gaussian process [47] on those labeled examples. Comparing the top two panels, we see that $\mathcal{D}_{\text{train}}$ (middle) contains an unbiased sample of the data in $\mathcal{D}_{\text{pool}}$ (top). However, in the bottom panel, we see that while the CATE estimator is accurate and confident near the modes of $\mathcal{D}_{\text{pool}}$, it becomes less accurate as we move to lower-density regions. In this way, the random acquisition of data reflects the biases inherent in $\mathcal{D}_{\text{pool}}$ and over-allocates resources to the modes of the distribution. If the mode were to coincide with a region of non-overlap, the function would most frequently acquire uninformative examples.

Next, we look at using the propensity score to bias data acquisition toward regions where the overlap assumption is satisfied.

**Definition 3.1.** *Counterfactual Propensity Acquisition*

$$\mathrm{I}(\widehat{\pi}_{\text{t}} \mid \mathbf{x}, \text{t}, \mathcal{D}_{\text{train}}) \equiv 1 - \widehat{\pi}_{\text{t}}(\mathbf{x}) \tag{4}$$

Intuitively, this function prefers points where the propensity for observing the counterfactual is high. We are considering the setup where $\mathcal{D}_{\text{pool}}$ contains observations of both $\mathbf{X}$ and $\mathrm{T}$, so it is straightforward to train an estimator for the propensity, $\widehat{\pi}_{\text{t}}(\mathbf{x})$. Figure 2b shows that while propensity score acquisition matches the treated and control densities in the train set, it still biases data selection towards the modes of $\mathcal{D}_{\text{pool}}$.

The goal of BALD is to acquire data $(\mathbf{x}, \text{t})$ that maximally reduces uncertainty in the model parameters $\mathbf{\Omega}$ used to predict the treatment effect. The most direct way to apply BALD is to use our uncertainty over the predicted treatment effect, expressed using the following information theoretic quantity:

**Definition 3.2.** *τBALD*

$$\mathrm{I}(\mathrm{Y}^1 - \mathrm{Y}^0; \mathbf{\Omega} \mid \mathbf{x}, \text{t}, \mathcal{D}_{\text{train}}) \approx \underset{\boldsymbol{\omega} \sim p(\mathbf{\Omega}|\mathcal{D}_{\text{train}})}{\mathrm{Var}} \left( \widehat{\mu}_{\boldsymbol{\omega}}(\mathbf{x}, 1) - \widehat{\mu}_{\boldsymbol{\omega}}(\mathbf{x}, 0) \right). \tag{5}$$

Building off the result in [20], we show how the LHS measure about the *unobservable potential outcomes* is estimated by the variance over $\mathbf{\Omega}$ of the *identifiable difference in expected outcomes* in Theorem 1 of the appendix. Alaa and van der Schaar [4] propose a similar result is for non-parametric models. Intuitively, this measure represents the information gain for $\mathbf{\Omega}$ if we observe the difference in potential outcomes $\mathrm{Y}^1 - \mathrm{Y}^0$ for a given measurement $\mathbf{x}$ and $\mathcal{D}_{\text{train}}$.

However, a fundamental flaw with this measure exists: observing labels for the random variable $\mathrm{Y}^1 - \mathrm{Y}^0$ is impossible. Thus, τBALD represents an irreducible measure of uncertainty. That is, τBALD will be high if it is uncertain about the label given the unobserved treatment $\text{t}'$, regardless of its certainty about the outcome given the factual treatment $\text{t}$, which makes τBALD highest for low-density regions and regions with no overlap. Figure 2c illustrates these consequences. We see the acquisition biases the training data away from the modes of the $\mathcal{D}_{\text{pool}}$, where we cannot know the treatment effect (no overlap). In datasets where we have limited overlap, it leads to uninformative acquisitions.

One remedy to the issues of τBALD is to only focus on reducible uncertainty:

**Definition 3.3.** *μBALD*

$$\mathrm{I}(\mathrm{Y}^{\text{t}}; \mathbf{\Omega} \mid \mathbf{x}, \text{t}, \mathcal{D}_{\text{train}}) \approx \underset{\boldsymbol{\omega} \sim p(\mathbf{\Omega}|\mathcal{D}_{\text{train}})}{\mathrm{Var}} \left( \widehat{\mu}_{\boldsymbol{\omega}}(\mathbf{x}, \text{t}) \right). \tag{6}$$

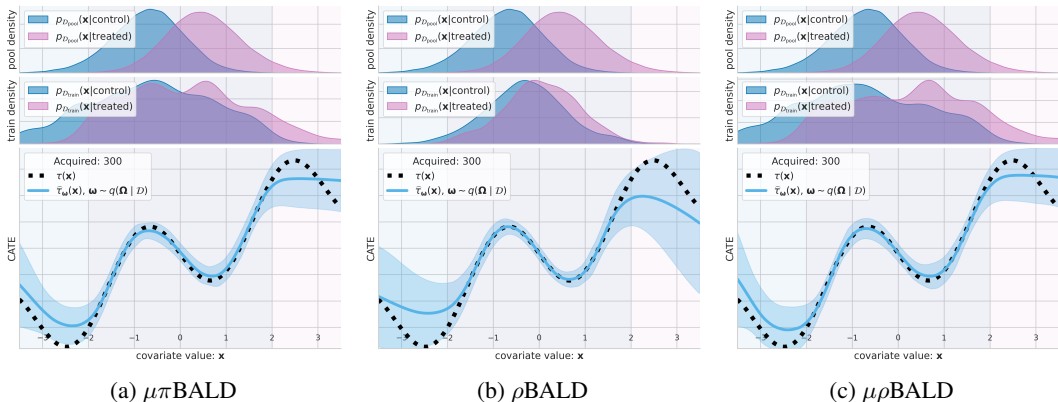

(a) $\mu\pi$BALD        (b) $\rho$BALD        (c) $\mu\rho$BALD

Figure 3: Causal–BALD acquisition functions: How the training set is biased and how this effects the CATE function with a fixed budget of 300 acquired points.

This measure represents the information gain for the model parameters $\mathbf{\Omega}$ if we obtain a label for the observed potential outcome $Y^t$ given a data point $(\mathbf{x}, t)$ and $\mathcal{D}_{\text{train}}$. We give proof for these results in Theorem 2 of the appendix.

$\mu$BALD only contains observable quantities; however, it does not account for our belief about the counterfactual outcome. As illustrated in Fig. 2d, this approach can prefer acquiring $(\mathbf{x}, t)$ when we are also very uncertain about $(\mathbf{x}, t')$, even if $(\mathbf{x}, t')$ is not in $\mathcal{D}_{\text{pool}}$. Since we can neither reduce uncertainty over such $(\mathbf{x}, t')$ nor know the treatment effect, the acquisition function would not be optimally data efficient.

## 3.2 Causal–BALD.

In the previous section, we looked at naive methods that either considered overlap or considered information gain. In this section, we present three measures that account for both factors when choosing a new point to acquire for model training.

A straightforward to combine knowledge about a data point's information gain and overlap is to simply multiply $\mu$BALD(6) by the propensity acquisition term (4):

**Definition 3.4.** $\mu\pi BALD$

$$\mathrm{I}(\mu\pi \mid \mathbf{x}, t, \mathcal{D}_{\text{train}}) \equiv (1 - \widehat{\pi}_t(\mathbf{x})) \underset{\boldsymbol{\omega} \sim p(\mathbf{\Omega}|\mathcal{D}_{\text{train}})}{\mathrm{Var}} (\widehat{\mu}_{\boldsymbol{\omega}}(\mathbf{x}, t)) \tag{7}$$

We can see in Fig. 3a that the acquisition of training data results in matched sampling that we saw for propensity acquisition in Fig. 2b. However, the tails of the overlapping distributions extend further into the low-density regions of the pool set support where the overlap assumption is satisfied.

Alternatively, we can take an information-theoretic approach to combine knowledge about a data point's information gain and overlap. Let $\widehat{\mu}_{\boldsymbol{\omega}}(\mathbf{x}, t)$ be an instance of the random variable $\widehat{\mu}_{\mathbf{\Omega}}^t \in \mathbb{R}$ corresponding to the expected outcome conditioned on $t$. Further, let $\widehat{\tau}_{\boldsymbol{\omega}}(\mathbf{x})$ be an instance of the random variable $\widehat{\tau}_{\mathbf{\Omega}} = \widehat{\mu}_{\mathbf{\Omega}}^1 - \widehat{\mu}_{\mathbf{\Omega}}^0$ corresponding to the CATE. Then,

**Definition 3.5.** $\rho BALD$

$$\mathrm{I}(Y^t; \widehat{\tau}_{\mathbf{\Omega}} \mid \mathbf{x}, t, \mathcal{D}_{\text{train}}) \gtrapprox \frac{1}{2} \log \left( \frac{\mathrm{Var}_{\boldsymbol{\omega}}(\widehat{\mu}_{\boldsymbol{\omega}}(\mathbf{x}, t)) - 2\,\mathrm{Cov}_{\boldsymbol{\omega}}(\widehat{\mu}_{\boldsymbol{\omega}}(\mathbf{x}, t), \widehat{\mu}_{\boldsymbol{\omega}}(\mathbf{x}, t'))}{\mathrm{Var}_{\boldsymbol{\omega}}(\widehat{\mu}_{\boldsymbol{\omega}}(\mathbf{x}, t'))} + 1 \right). \tag{8}$$

This measure represents the information gain for the CATE $\tau_{\mathbf{\Omega}}$ if we observe the outcome $Y$ for a datapoint $(\mathbf{x}, t)$ and the data we have trained on $\mathcal{D}_{\text{train}}$. We give proof for this result in Theorem 3.

In contrast to $\mu$-BALD, this measure accounts for overlap in two ways. First, $\rho$–BALD will be scaled by the inverse of the variance of the expected counterfactual outcome $\widehat{\mu}_{\boldsymbol{\omega}}(\mathbf{x}, t')$. This scaling biases acquisition towards examples for which we know about the counterfactual outcome, so we can assume that overlap is satisfied for observed $(\mathbf{x}, t)$. Second, $\rho$–BALD is discounted by $\mathrm{Cov}_{\boldsymbol{\omega}}(\widehat{\mu}_{\boldsymbol{\omega}}(\mathbf{x}, t), \widehat{\mu}_{\boldsymbol{\omega}}(\mathbf{x}, t'))$. This discounting is a concept that we will leave for future discussion.

In Fig. 3b we see that $\rho$–BALD has matched the distributions of the treated and control groups similarly to propensity acquisition in Fig. 2b. Further, we see that the CATE estimator is more accurate over the support of the data.

There is, however, a shortcoming of $\rho$–BALD that may lead to suboptimal data efficiency. Consider two examples in $\mathcal{D}_{\text{pool}}$, $(\mathbf{x}_1, t_1)$ and $(\mathbf{x}_2, t_2)$ where $\text{Var}_{\boldsymbol{\omega}}(\widehat{\mu}_{\boldsymbol{\omega}}(\mathbf{x}_1, t_1)) = \text{Var}_{\boldsymbol{\omega}}(\widehat{\mu}_{\boldsymbol{\omega}}(\mathbf{x}_1, t_1'))$ and $\text{Var}_{\boldsymbol{\omega}}(\widehat{\mu}_{\boldsymbol{\omega}}(\mathbf{x}_2, t_2)) = \text{Var}_{\boldsymbol{\omega}}(\widehat{\mu}_{\boldsymbol{\omega}}(\mathbf{x}_2, t_2'))$: for each point, we are as uncertain about the conditional expectation given the factual treatment as we are uncertain given the counterfactual treatment. Further, let $\text{Cov}_{\boldsymbol{\omega}}(\widehat{\mu}_{\boldsymbol{\omega}}(\mathbf{x}_1, t_1), \widehat{\mu}_{\boldsymbol{\omega}}(\mathbf{x}_1, t_1')) = \text{Cov}_{\boldsymbol{\omega}}(\widehat{\mu}_{\boldsymbol{\omega}}(\mathbf{x}_2, t_2), \widehat{\mu}_{\boldsymbol{\omega}}(\mathbf{x}_2, t_2'))$. Finally, let $\text{Var}_{\boldsymbol{\omega}}(\widehat{\mu}_{\boldsymbol{\omega}}(\mathbf{x}_1, t_1)) > \text{Var}_{\boldsymbol{\omega}}(\widehat{\mu}_{\boldsymbol{\omega}}(\mathbf{x}_2, t_2))$: we are more uncertain about the conditional expectation given the factual treatment for data point $(\mathbf{x}_1, t_1)$ than we are for data point $(\mathbf{x}_2, t_2)$. Under these three conditions, $\rho$–BALD would rank these two points equally, and so this method would bias training data to the modes of $\mathcal{D}_{\text{pool}}$ when $(\mathbf{x}_2, t_2)$ is more frequent than $(\mathbf{x}_1, t_1)$. In practice, it may be more data-efficient to choose $(\mathbf{x}_1, t_1)$ over $(\mathbf{x}_2, t_2)$ as it would more likely be a point as yet unseen by the model.

To combine the positive attributes of $\mu$–BALD and $\rho$–BALD, while mitigating their shortcomings, we introduce $\mu\rho$BALD.

**Definition 3.6.** $\mu\rho BALD$

$$\text{I}(\mu\rho \mid \mathbf{x}, t, \mathcal{D}_{\text{train}}) \equiv \underset{\boldsymbol{\omega}}{\text{Var}}\left(\widehat{\mu}_{\boldsymbol{\omega}}(\mathbf{x}, t)\right) \frac{\text{Var}_{\boldsymbol{\omega}}(\widehat{\tau}_{\boldsymbol{\omega}}(\mathbf{x}))}{\text{Var}_{\boldsymbol{\omega}}(\widehat{\mu}_{\boldsymbol{\omega}}(\mathbf{x}, t'))}. \tag{9}$$

Here, we scale Equation 8, which has equivalent expression $\frac{\text{Var}_{\boldsymbol{\omega}}(\widehat{\tau}_{\boldsymbol{\omega}}(\mathbf{x}))}{\text{Var}_{\boldsymbol{\omega}}(\widehat{\mu}_{\boldsymbol{\omega}}(\mathbf{x}, t'))}$ by our measure for $\mu$BALD such that in the cases where the ratio may be equal, there is a preference for data points the current model is more uncertain about. We can see in Fig. 3c that the training data acquisition is distributed more uniformly over the support of the pool data where the overlap assumption is satisfied. Furthermore, the accuracy of the CATE estimator is highest over that region.

## 4    Related Work

Deng et al. [10] propose the use of Active Learning for recruiting patients to assign treatments that will reduce the uncertainty of an Individual Treatment Effect model. However, their setting is different from ours – we assume that suggesting treatments are too risky or even potentially lethal. Instead, we acquire patients to reveal their outcome (e.g., by having a biopsy). Additionally, although their method uses predictive uncertainty to identify which patients to recruit, it does not disentangle the sources of uncertainty; therefore, it will also recruit patients with high outcome variance. Closer to our proposal is the work from Sundin et al. [45]. They propose using a Gaussian process (GP) to model the individual treatment effect and use the expected information gain over the S-type error rate, defined as the error in predicting the sign of the CATE, as their acquisition function. Although GPs are suitable for quantifying uncertainty, they do not work well on high-dimensional input spaces. In this work, we use Neural network methods to obtain uncertainty: Deep Ensembles [28] and DUE [47], a Deep Kernel Learning GP, both of which work well even on high dimensional inputs. Additionally, the authors assume that noisy observations about the counterfactual treatments are available at training time where we make no such assumptions. We compare to this in our experiment by limiting the access to counterfactual observations ($\gamma$ baseline) and adapting it to Deep Ensembles [28] and DUE [47] (we provide more details about the adaptation in Appendix B.1). Recent work by Qin et al. [39] looks at budgeted heterogeneous effect estimation but does not factor weak or limited overlap into their acquisition function.

# 5 Experiments

In this section, we evaluate our acquisition objectives on synthetic and semi-synthetic datasets. Code to reproduce these experiments is available at https://github.com/anndvision/causal-bald.

**Datasets** Starting from the hypothesis that different objectives can target different types of imbalances and degrees overlap, we construct a **synthetic** dataset [22] demonstrating the various biases. We depict this dataset graphically in Fig. 1. We use this dataset primarily for illustrative purposes. By design, we have constructed a primary data mode and have regions of weak or no overlap. Additionally, we study the performance of our acquisition functions on the **IHDP** dataset [17, 42], which is a standard benchmark in causal treatment effect literature. Finally, we demonstrate that our method is suitable for high dimensional, large-sample datasets on **CMNIST** [21], an MNIST [29] based dataset adapted for causal treatment effect studies. In Fig. 4, we see that CMNIST is an adaptation of the synthetic dataset. Model inputs are MNIST digits and assigned treatments, and the response surfaces are generated based on a projection of the digits onto a latent 1-dimensional manifold. The observed digits are high-dimensional proxies for the confounding covariate $\phi$. Detailed descriptions of each dataset are available in Appendix C.

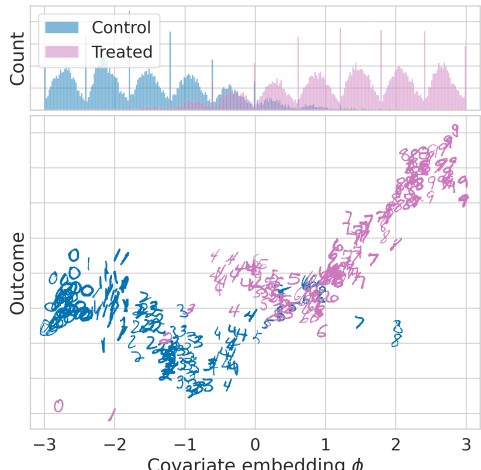

Figure 4: Visualizing CMNIST dataset. Model inputs are MNIST digits and assigned treatments. The MNIST digits are high-dimensional proxies for the latent confounding covariate $\phi$. Digits are projected onto $\phi$ by ordering them first by image intensity and then by digit class (0 - 9). Methods must be able to implicitly learn this non-linear mapping in order to predict the conditional expected outcomes.

**Model** Our objectives rely on methods that are capable of modeling uncertainty and handling high-dimensional data modalities. DUE [47] is an instance of Deep Kernel Learning [48] that uses a deep feature extractor to transform the inputs and defines a Gaussian process (GP) kernel over the extracted feature representation. In particular, DUE uses a variational inducing point approximation [16] and a constrained feature extractor that contains residual connections and spectral normalization to enable reliable uncertainty. Due obtains SotA results on IHDP [47]. In DUE, we distinguish between the model parameters $\theta$ and the variational parameters $\omega$, and we are Bayesian only over the $\omega$ parameters. Since DUE is a GP, we obtain a full Gaussian posterior over outcomes from which we can use the mean and covariance directly. When necessary, sampling is very efficient and only requires a single forward pass in the deep model. We describe all hyperparameters in Appendix F.

**Baselines** We compare against the following baselines: **Random.** This acquisition function selects points uniformly at random. **Propensity.** An acquisition function based on the propensity score (Eq. 4). We train a propensity model on the pool data, which we then use to acquire points based on their propensity score. Please note that this is a valid assumption as training a propensity model does not require outcomes. $\gamma$ **(S-type error rate) [45].** This acquisition function is the S-type error rate based method proposed by Sundin et al. [45]. We have adapted the acquisition function to use with Bayesian Deep Neural Networks. The objective is defined as $I(\gamma; \Omega \mid \mathbf{x}, \mathcal{D}_{\text{train}})$, where $\gamma(x) = \text{probit}^{-1}\left(-\frac{|\mathbb{E}_{p(\tau|\mathbf{x}, \mathcal{D}_{\text{train}})}[\tau]|}{\sqrt{\text{Var}(\tau|\mathbf{x}, \mathcal{D}_{\text{train}})}}\right)$ and $\text{probit}^{-1}(\cdot)$ is the cumulative distribution function of normal distribution. In contrast to the original formulation, we do not assume access to counterfactual observations at training time.

## 5.1 Experimental Results

For each of the acquisition objectives, dataset, and model we present the mean and standard error of empirical square root of precision in estimation of heterogenous effect (PEHE) [2]. We summarize

---

[2] $\sqrt{\epsilon_{PEHE}} = \sqrt{\frac{1}{N}\sum_x \left(\hat{\tau}(x) - \tau(x)\right)^2}$

Table 1: Summary of active learning parameters for each dataset.

| Dataset | Warm-up size | Acquisition size | Acquisition steps | Pool Size | Valid Size |
|---|---|---|---|---|---|
| Synthetic | 10 | 10 | 30 | 10k | 1k |
| IHDP | 100 | 10 | 38 | 471 | 201 |
| CMNIST | 250 | 50 | 55 | 35k | 15k |

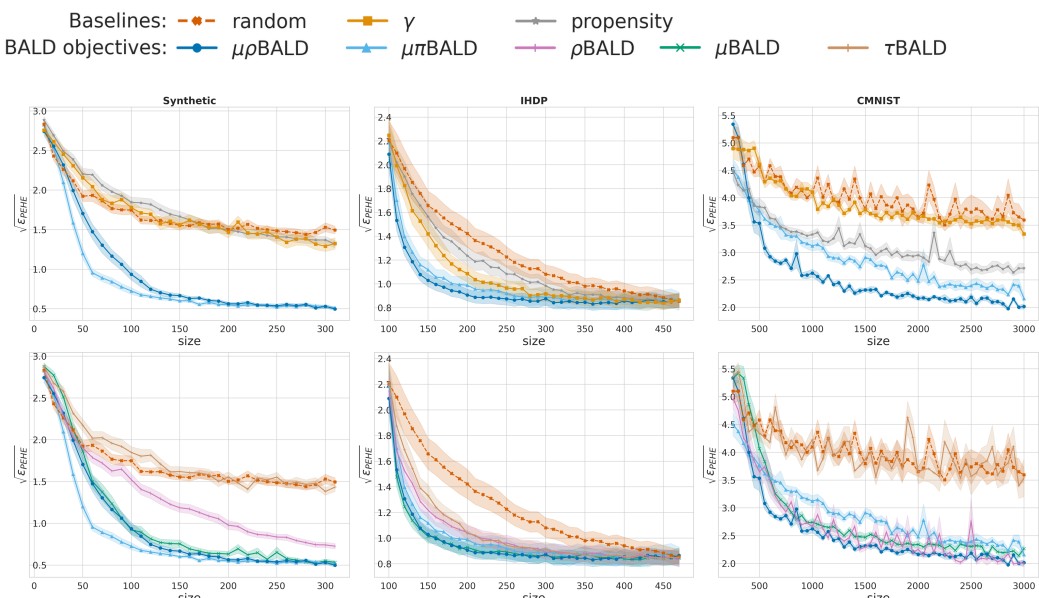

Figure 5: $\sqrt{\epsilon_{PEHE}}$ performance (shaded standard error) for DUE models. **(left to right) synthetic** (40 seeds), and **IHDP** (200 seeds). We observe that BALD objectives outperform the **random**, $\gamma$ and **propensity** acquisition functions significantly, suggesting that epistemic uncertainty aware methods that target reducible uncertainty can be more sample efficient.

each active learning setup in Table 1. The *warm up size* is the number of examples in the initial pool dataset. *Acquisition size* is the number of examples labeled at each acquisition step. *Acquisition steps* is the number of times we query a batch of labels. *Pool size* is the number of examples in the pool dataset. Finally, *valid size* is the number of examples used for model selection when optimizing the model at each acquisition step.

In Fig. 5, we see that epistemic uncertainty aware $\mu\rho$BALD outperforms the baselines, random, propensity, and S-Type error rate ($\gamma$). As analyzed in section 3, we expect this improvement as our acquisition objectives target reducible uncertainty – that is, epistemic uncertainty when there is overlap between treatment and control. Additionally, $\mu\rho$BALDshows superior performance over the other objectives in the high dimensional dataset CMNIST verifying our qualitative analysis in Figure 3c.

Each of ($\mu$BALD, $\rho$BALD, $\mu\pi$BALD, and $\mu\rho$BALD) outperform the baseline methods on these tasks. Of note, the performance $\rho$BALD improves as the dimensionality of the covariates increases. In contrast, the performance of the propensity score-based $\mu\pi$BALD worsens as the dimensionality of the covariates increases. Propensity score estimation is known to be a problem in high-dimensions [12]. We see that both $\mu$BALD and $\mu\rho$BALD perform consistently as dimensionality increases, with $\mu\rho$BALD showing a statistically significant improvement over $\mu$BALD on two of the three tasks. These improvements indicate that $\mu\rho$BALD is more robust for data with high-dimensional covariates than $\mu\pi$BALD ; moreover, $\mu\rho$BALD does not need an additional propensity score model.

## 6   Conclusion

We have introduced a new acquisition function for active learning of individual-level causal-treatment effects from high dimensional observational data, based on Bayesian Active Learning by Disagreement [18]. We derive our proposed method from an information-theoretic perspective and compare it with acquisition strategies that either do not consider epistemic uncertainty (i.e., random or propensity-based) or target irreducible uncertainty in the observational setting (i.e., when we do not have access to counterfactual observations). We show that our methods significantly outperform baselines, while also studying the various properties of each of our proposed objectives in both quantitative and qualitative analyses, potentially impacting areas like healthcare where sample efficiency in the acquisition of new examples implies improved safety and reductions in costs.

## 7   Broader Impact

Active Learning for learning treatment effects from observational data is highly actionable research, and there are several sectors where our research can have an impact. Take, for example, a hospital that needs to decide whom to treat, based on some model. To these choices, the decision-maker needs to have a confident and accurate treatment effect prediction model. However, improving the performance of such a model requires data from patients, which might be costly and perhaps even unethical to acquire. With this work, we assume that we cannot assign new treatments to patients but only perform biopsies or questionnaires post-treatment to reveal the outcome. We believe that this is an impactful and realistic scenario that will directly benefit from our proposal. However, our method can also impact fields like computational-advertisement, where the goal is to learn a model to predict the captivate the attention of users, or policymaking where a government wants to decide how to intervene for beneficial or malicious reasons.

Active learning inherently biases the acquisition of training data. We attempt to show how this can be beneficial or detrimental for learning treatment effects under different acquisition functions under the unconfoundedness assumption. We are not making guarantees on the overall unbiasedness of our methods. We guarantee only that our results are conditional on the unconfoundedness assumption. Unobserved confounding can result in the biased sampling of training data concerning the hidden confounding variable. This bias can result in performance inequality between groups and a biased estimate of the unconfounded CATE function. Further, the models' uncertainty estimates are not informative of when this may occur.

An anonymous reviewer writes, "one risk is that the method could, e.g., lead to a biased, non-representative sampling in terms of ethnicity and other protected attributes - particularly, if the unconfoundedness assumption this work is based on is blindly trusted" Recent work by Andrus et al. [5] does a great job of highlighting the difficulties practitioners face when accounting for algorithmic bias across protected attributes. In such cases, model uncertainty is not enough to identify non-representative sampling concerning the protected attribute. They report about a practitioner's use of structural causal models in concert with domain expert feedback as a means to inform clients of potential sources of bias. Perhaps such methodology could be used with causal sensitivity analysis for CATE [22, 50, 21] as a way to model beliefs about the protected attribute without observing it.

Impactful avenues for future work include relaxing the unconfoundedness assumption, incorporating beliefs about hidden confounding into the acquisition function. Furthermore, in addition to uncovering the outcome, we think it is interesting to revisit the active treatment assignment problem. On the active learning side, exploring more rigorous batch acquisition methods could yield improvements over the current stochastic sampling estimation we use. Finally, in this work, we assume access to a validation set, which may not be available in practice, so exploration of active acquisition of validation data will also have an impact [27].

## Acknowledgments and Disclosure of Funding

We would like to thank OATML group members and all anonymous reviewers for sharing their valuable feedback and insights. PT and AK are supported by the UK EPSRC CDT in Autonomous Intelligent Machines and Systems (grant reference EP/L015897/1). JvA is grateful for funding by the EPSRC (grant reference EP/N509711/1) and by Google-DeepMind. U.S. was partially supported by the Israel Science Foundation (grant No. 1950/19).

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
