# OpenReview forum: "Causal-BALD: Deep Bayesian Active Learning of Outcomes to Infer Treatment-Effects from Observational Data"
_NeurIPS.cc/2021/Conference — NeurIPS 2021 Poster_

### Official Review · Reviewer_8K1A · 2021-07-04

**Rating:** 8
**Confidence:** 4

**Summary:**

Update (12th Aug 2021) - increased score to reflect that my comments were satisfactorily addressed by the authors in their rebuttal.

-----------
The authors propose new set of acquisition functions, dubbed Causal-BALD, based on Bayesian Active Learning by Disagreement (BALD) that consider overlap and information gain as a decision criterion for iterative improvement of a counterfactual estimator in settings in which direct treatment assignment by an algorithm is unethical or not feasible. The authors performed a clean evaluation in various settings and with multiple underlying models to show that their proposed approach outperforms the previous state-of-the-art.

**Ethical Concerns:**

I have no ethical concerns.

**Limitations And Societal Impact:**


**Societal Impact**:
- As a general-purpose method for active learning in the counterfactual estimation setting with potentially broad applicability, I see no immediate reason for additional concern beyond that inherent in any active learning method. One risk is that the method could, e.g., lead to a biased, non-representative sampling in terms of ethnicity and other protected attributes - particularly, if the unconfoundedness assumption this work is based on is blindly trusted. The authors could briefly outline this risk and how one could preempt such a scenario with appropriate checks and balances.

**Questions/suggestions for the authors**:
- Please describe how the experimental hyper-parameters presented in the supplementary materials were chosen. In general, conducting experiments over multiple hyper-parameter sets would be preferable to ensure the results generalize to other choices of hyper-parameters for the underlying models.

- From the appendix, it appears the authors used a variant of TARNet [1] as constituents of the deep ensemble models (at least in some experiments). However, as far as I can see, [1] is not disclosed/referenced anywhere in the paper, nor is there a discussion of deep counterfactual representation learning methods in the paper. Please add the missing information on what models were the constituents of the deep ensembles and any related discussion to the related work section.

- Please include a short discussion on how the [standard in the field but relatively strict] assumptions (particularly the unconfoundedness assumption) and its potential impact - if violated - on the uncertainty estimates may affect your proposed acquisition function in practice so that readers are aware of this inherent risk.

- Please add treatment-naive BALD (eq.s 1-2) to the experimental comparison to highlight and quantify the benefit of adapting the acquisition function to the counterfactual setting.

**Conclusion**: A relevant and understudied setting, clear and compelling narrative and a comprehensive experimental evaluation. There are minor issues that I trust can be resolved during review. This paper ticks all the boxes for a clear accept.

Note: I have not verified the proofs in the supplementary materials.

[1] Shalit, Uri, Fredrik D. Johansson, and David Sontag. "Estimating individual treatment effect: generalization bounds and algorithms." International Conference on Machine Learning. PMLR, 2017.


**Main Review:**

**Originality**:
- The authors study the application of active learning methods for interactively querying a data set for the improvement of an underlying counterfactual estimator (deep ensembles and DUE). To the best of my knowledge, this is a non-trivial, relevant and understudied setting, and originality is therefore clearly a strength of the presented work.

**Quality**:
- The performed experiments and their descriptions are comprehensive. The authors evaluate multiple relevant settings, underlying models and benchmark datasets to substantiate their claims.

- The benefits of Causal-BALD over treatment-naive BALD (eq.s 1-2) are discussed at length in the manuscript. However, a treatment-naive BALD baseline is missing from the experimental comparison.

**Clarity**:
- The paper is well written and the text is easy to follow for a reader versed in the field of counterfactual inference. The exposition is clear and the paper follows a compelling narrative.

- How experimental hyper-parameters were systematically chosen for the models is not described in the manuscript. Only the final hyper-parameters used in the experiments are presented in the supplementary materials. On review of the source code, I saw that a systematic approach was likely performed (see tuning.py) - this information should be included in the manuscript.

- In terms of reproducibility, I particularly appreciate the comprehensive appendix that covers the details missing in the main body of the paper and the clear reference implementation provided by the authors.

**Significance**:
- Active learning for estimating individual treatment effects is an understudied, but relevant, problem with several potential practical applications. The proposed method seems to be generally applicable under the stated assumptions, and the experimental results support its use.

- The presented acquisition functions, crucially, rely on the availability of meaningful uncertainty estimates. An additional source of uncertainty in the counterfactual estimation setting is the potential presence of unobserved confounding that can not be ruled out in practical scenarios and will not be reflected in the models' uncertainty estimates. Claims towards "unbiasedness" should therefore be toned down to reflect that we can not test the unconfoundedness assumption, and the proposed method therefore will not have guarantees to be unbiased in  real-world applications.


**Time Spent Reviewing:**

4

---

> ### Author Response · Authors · 2021-08-10
> **Response to Reviewer 8K1A**
>
> We thank the reviewer for their valuable feedback. We address their comments on hyper-parameter selection, deep ensemble details, treatment-naive BALD, and hidden confounding in the following.
>
> **“How experimental hyper-parameters were systematically chosen for the models is not described in the manuscript. Only the final hyper-parameters used in the experiments are presented in the supplementary materials. On review of the source code, I saw that a systematic approach was likely performed (see tuning.py) - this information should be included in the manuscript.”**
>
> Thank you for taking the time and reading the code as well. We will update the main text explaining that we use the hyperopt tuning strategy while including the hyper-parameter space we used. This ought to be a simple summary of the code you saw in tuning.py. We appreciate the time you took to go beyond the main text to locate this information.
>
> **“Please add the missing information on what models were the constituents of the deep ensembles and any related discussion to the related work section.”**
>
> Thank you for this suggestion. We will update the main text explaining the base model (TARNET) that we used in our experiments. We will further expand the backgrounds and related works sections to include references to deep ensembles.
>
> **“Please add treatment-naive BALD (eq.s 1-2) to the experimental comparison to highlight and quantify the benefit of adapting the acquisition function to the counterfactual setting.”**
>
> We would like to direct the reviewer’s attention to $\tau-BALD$. Is this what you refer to as treatment-naive BALD? As you can see in the experiments section, this baseline is outperformed by causal-BALD objectives.
>
> **“An additional source of uncertainty in the counterfactual estimation setting is the potential presence of unobserved confounding that can not be ruled out in practical scenarios and will not be reflected in the models' uncertainty estimates. Claims towards "unbiasedness" should therefore be toned down to reflect that we can not test the unconfoundedness assumption, and the proposed method therefore will not have guarantees to be unbiased in real-world applications.”**
>
> This is a very good point. You are absolutely correct that unobserved confounding can result in biased sampling of training data with respect to the hidden confounding variable. This can result in both performance inequality between groups and a biased estimate of the unconfounded CATE function. Further, we agree that the models’ uncertainty estimates are not informative of when this may occur. However, we only claim that random sampling results in an unbiased sample of training data with respect to the pool set distribution. We then highlight how this can be problematic even when unconfoundedness holds. Active learning inherently biases the acquisition of training data, and we try to show how this can be beneficial or detrimental for learning treatment effects under different acquisition functions, under the unconfoundedness assumption. We will clarify that we are not making guarantees on the overall unbiasedness of our methods, but rather that our guarantees are conditional on the unconfoundedness assumption. We will include a discussion on how hidden confounding can lead to biased representation and performance in the space of unobserved variables as well as a biased estimate of the true CATE function.
>
> **“One risk is that the method could, e.g., lead to a biased, non-representative sampling in terms of ethnicity and other protected attributes - particularly, if the unconfoundedness assumption this work is based on is blindly trusted. The authors could briefly outline this risk and how one could preempt such a scenario with appropriate checks and balances.”**
>
> This is another excellent point. Recent work by [1] does a great job of highlighting the difficulties practitioners face when accounting for algorithmic bias across protected attributes. In such cases, model uncertainty is not enough to identify non-representative sampling with respect to the protected attribute. They report about a practitioner's use of structural causal models in concert with domain experts as a way to inform clients of potential sources of bias. Perhaps such methodology could be used with causal sensitivity analysis for CATE  [2, 3] as a way to model beliefs about the protected attribute without observing it. We would be happy to add commentary on this problem, and amplify similar works to bring greater attention to this very real concern. How to properly incorporate this into the active learning process is a definite avenue of future research.
>
> [1] Andrus, McKane, et al. "What We Can't Measure, We Can't Understand: Challenges to Demographic Data Procurement in the Pursuit of Fairness." Proceedings of the 2021 ACM Conference on Fairness, Accountability, and Transparency. 2021.
>
> [2] Yadlowsky, Steve, et al. "Bounds on the conditional and average treatment effect with unobserved confounding factors." arXiv preprint arXiv:1808.09521 (2018).
>
> [3] Kallus, Nathan, Xiaojie Mao, and Angela Zhou. "Interval estimation of individual-level causal effects under unobserved confounding." The 22nd international conference on artificial intelligence and statistics. PMLR, 2019.

---

> > ### Comment · Reviewer_8K1A · 2021-08-12
> > **Response to rebuttal**
> >
> > Many thanks for providing a detailed and comprehensive response.
> > I have updated my score to reflect that my questions were satisfactorily addressed by the authors.

---

> > > ### Author Response · Authors · 2021-08-20
> > > **Thank you**
> > >
> > > Thank you again for your valuable feedback and for your generous scoring of our work.

---

### Official Review · Reviewer_wzW6 · 2021-07-16

**Rating:** 6
**Confidence:** 3

**Summary:**

This paper proposed a new acquisition function for applying active learning to observation data consisting of covariates, treatments and outcomes. The motivation of developing a new acquisition function is the existing ones could either query too many samples close to the distribution mode or query samples outside the overlapping region of the treatment and control group. Therefore, the key contribution of this work is to propose a acquisition function that 1) query samples in the overlapping region of case and control; and 2) reduce uncertainity of the model parameters.

The proposed function is built on the existing Bayesian active learning by disagreement (BALD).

First, \tauBALD aims to maximize the conditional mutual information between CATE and model parameters \Omega. However, since the control outcome and treatment outcome cannot be observed simultaneously, leading to irreducible uncertainity.

Second, to overcome this drawback, \muBALD was considered by maximizing the conditional mutual information between outcome Y^t and the model parameters \Omega. The proposed Causal-BALD improved \muBALD by considering both model uncertainty reduction and overlapping. Specifically, \mu\piBALD directly multiplies the propensity acquisition function and \muBALD.

Third, \rhoBALD can also handle overlap by maximizing the mutual information between outcome Y^t and the CATE computed from model parameter \Omega. \rhoBALD can also be combined to \muBALD to form \mu\rhoBALD, which can handle both model uncertainity reduction and overlap.

Experimental results on synthetic dataset and semi-synthetic data showed the proposed \mu\rhoBALD outperforms existing acquisition function for the application active learning to individual treatment effect estimation.

**Ethics Review Area:**

["I don’t know"]

**Main Review:**

This paper is well motivated. The problem setting of treatment effect estimation requires an adapted version of active learning algorithms that can focus on acquire samples from the overlap region. This work started from this intuition and proposed new acquisition function that can achieve both overlap and uncertainty reduction. The experimental results show the performance gain of the new acquisition function compared to existing baseline approaches.

I don't have major complains on the method of this work. A few minor comments are listed below.

1) A few variations of BALD are listed in this work. It could help to highlight the major contribution of this work in the abstract or introduction so that readers can clearly grasp this work's novelty.

2) In Table 2, are these results based on within-sample evaluation or out-of-sample evaluation?

3) Did the authors try to run experiments on other commonly used benchmark datasets for treatment effect estimation, such as Twins and Jobs?

4) The font size in the legend in Figure 2 should be increased to become more readable.

**Time Spent Reviewing:**

1.5

---

> ### Author Response · Authors · 2021-08-10
> **Response to Reviewer wzW6**
>
> We thank the reviewer for their valuable feedback. We address their comments on highlighting major contributions, out-of-sample evaluation, benchmark datasets, and readability in the following.
>
> **“A few variations of BALD are listed in this work. It could help to highlight the major contribution of this work in the abstract or introduction so that readers can clearly grasp this work's novelty.”**
>
> This is indeed a great suggestion and we will update the introduction with a clear statement about the novelty. We hope the sentence below succeeds on this goal:
>
> line 88: Our proposed objectives, termed "Causal BALD" ($\rho-BALD$, $\mu\rho-BALD$, $\mu\pi-BALD$) consider not only the information gain but also the overlap between treated and control population, whereas previous work has been mostly focused on information gain as the acquisition objective. It is crucial to select data points where overlap is satisfied as the CATE is not identifiable where it is not.
>
> We have also considered moving the discussion of $\mu\pi-BALD$ to the appendix and only reporting results for either ensemble, or DUE in the main text, while moving the other model to the appendix. This could free up space for an extended discussion on limitations as suggested by other reviewers, and potentially make the narrative/contributions easier to follow. We are interested in the reviewer’s thoughts about these potential changes.
>
> **“In Table 2, are these results based on within-sample evaluation or out-of-sample evaluation?”**
>
> We evaluate the results in a holdout (out-of-sample) dataset which aims to evaluate the generalization capabilities of the trained models.
>
> **“Did the authors try to run experiments on other commonly used benchmark datasets for treatment effect estimation, such as Twins and Jobs?”**
>
> We have not yet run experiments on Jobs or Twins. Given limited resources, we chose to focus on the synthetic example to illustrate the problem, the IHDP dataset to show results on a medium dimension benchmark dataset, and the CMNIST dataset to show that the model scales to large-sample, high-dimensional data. We think that the additional results you propose would add value for follow up research and we would be happy to include them in the camera ready version should this work be accepted.
>
> **“The font size in the legend in Figure 2 should be increased to become more readable.”**
>
> Thank you for the suggestion, we will increase the font size to improve readability and accessibility.

---

> > ### Comment · Reviewer_wzW6 · 2021-08-16
> > **Response to rebuttal**
> >
> > I would like to thank the authors for providing detailed responses to my questions. I have no further questions and would recommend the acceptance of this work.

---

> > > ### Author Response · Authors · 2021-08-20
> > > **Thank you**
> > >
> > > Thank you again for your valuable feedback and for your recommendation to accept.

---

### Official Review · Reviewer_EmX1 · 2021-07-16

**Rating:** 7
**Confidence:** 2

**Summary:**

This paper introduces "CausalBALD", a set of acquisition functions for use in Bayesian Active Learning when trying to estimate the Conditional Average Treatment Effect ("CATE") associated with an intervention, $\tau(\mathbf{x}) = \mathbb{E}[Y^{1}-Y^{0}|\mathbf{x}]$.  The authors introduce multiple information-theoretic acquisition functions, and then show the improved performance (relative to random sampling of examples) of their acquisition functions for estimating the true treatment effects present in three simulated/semi-synthetic datasets.  Through a toy problem (Figures 2/3), they also provide some intuition for why their acquisition functions outperform random sampling.

**Limitations And Societal Impact:**

The comments that I have relating to this paper are mostly minor:
- It would have been nice to have had a fuller description of the CMNIST dataset in text.  Could you even have shown a few examples of the labels, y, generated for a few example digits?  It was hard, if not impossible, to conceive of the elements of CMNIST from the description given in the main text.  The math in the supplement helped, but I am still having trouble visualizing exactly what some elements of this dataset look like.
- Do the authors have an explanation for why particular acquisition functions work better for some datasets than others?  E.g. it seems that $\mu\pi$-BALD works well on the synthetic dataset, but not so well on CMNIST
- It would also have been nice to have had (even in the supplement) a histogram of true treatment effects associated with each of the datasets used for evaluation.  I am wondering if your method works equally well for estimating all sizes of treatment effects, or if it is better at estimating larger treatment effects
- Finally, it would have been nice to have seen some analysis of the examples selected by the authors' best performing acquisition functions.
 This would have been particularly interesting in e.g. the CMNIST case, where one acquisition function vastly outperforms the other.  Which examples are being selected in that case, compared to for other acquisition functions?

**Main Review:**

The paper was well-written and clear in its exposition.  The problem that the authors aim to tackle is interesting and important.  In many scenarios where you wish to estimate a treatment effect, you are also limited in the amount of data you can acquire - hence, applying Bayesian active learning here seems very natural. Additionally, as far as I know, there is not a lot of literature on using active learning when estimating treatment effects (aside from the two papers that the authors cite - namely Deng et al. (2011) and Sundin (2019)).  The authors' approach seems reasonable, and I liked the fact that the authors evaluated the performance of their method on not just 1, but 3 datasets.

**Time Spent Reviewing:**

3

---

> ### Author Response · Authors · 2021-08-10
> **Response to Reviewer EmX1**
>
> We thank the reviewer for their valuable feedback. We address their comments on the CMNIST dataset, the performance of $\mu\pi-BALD$, performance across effect sizes, and analysis of acquired samples in the following.
>
> **“It would have been nice to have had a fuller description of the CMNIST dataset in text.”**
>
> Thank you for pointing out the confusing points about CMNIST. We will add this image https://imgur.com/tlvHKfE which hopefully clarifies how the CMNIST dataset looks like. The x axis enlists the MNIST digits and encodes them to different x values through the $\phi$ function, which maps digits [0, 9] to consecutive, non-overlapping subsets on the range (-2.5, 2.5). Each digit image is mapped onto it’s subset based on the image brightness. Then we artificially construct areas of high and low overlap (the shaded blue and orange regions represent the overlap). The dashed line visualizes the true treatment effect.
>
> **“Do the authors have an explanation for why particular acquisition functions work better for some datasets than others? E.g. it seems that μπ-BALD works well on the synthetic dataset, but not so well on CMNIST”**
>
> Thank you for this insightful question. $\mu\pi$-BALD follows the performance of the propensity model which we find gets worse as the dimensionality of the dataset increases. This trend can be seen by looking at the grey starred lines in the top rows of figures 5 and 6 of the appendix. We see the same trend for $\mu$-BALD which again worse with the dimensionality of the dataset. This trend can be seen by looking at the green x’ed lines in the second rows of figures 5 and 6 in the appendix. It seems like $\mu\pi$-BALD performs best when the propensity model and $\mu$-BALD work synergistically, while mostly influenced by the $\mu$-BALD component. We can observe that $\mu\pi$-BALD and $\mu$-BALD have similar performance on the high dimensional datasets but $\mu\pi$-BALD outperforms $\mu$-BALD on the synthetic dataset.
>
> We do use a deterministic propensity model, so perhaps it is making erroneous predictions in low density areas of the pool data. These errors could be propagating to the acquisition function making for suboptimal acquisitions. We will add an additional comment on the main text discussing these observations.
>
> **“It would also have been nice to have had (even in the supplement) a histogram of true treatment effects associated with each of the datasets used for evaluation. I am wondering if your method works equally well for estimating all sizes of treatment effects, or if it is better at estimating larger treatment effects"**
>
> Interesting question. This linked GIF https://imgur.com/MjdtzBy shows the progression of the true versus predicted CATE over each acquisition step using the $\mu\rho-BALD$ acquisition function for the IHDP dataset. We can see that it is slightly less accurate and more uncertain for outlier treatment effects. However, within the data distribution the method appears equally accurate across all effect sizes. Does this help address your question? We would be happy to add such visualizations for both IHDP and CMNIST to our public repo when the code is released.
>
> **“It would have been nice to have seen some analysis of the examples selected by the authors' best performing acquisition functions.”**
>
> We hope the following illustrations help give intuition for which examples are being selected at each acquisition step. They certainly helped us when developing the methods.
>
> On the left pane we show samples from the approximate posterior modeling the treatment effect. The right pane shows the acquisition score per datapoint according to which we sample and the history of acquired points (check this annotated frame https://imgur.com/a/XK4HjXH ).
>
> As you can see, [$\mu\rho-BALD$](https://imgur.com/a/9M7xmqd) converges to the true treatment effect with low uncertainty faster than [random](https://imgur.com/a/ThkfTDq) acquisition function. As you can see from the acquisition animation [$\mu\rho-BALD$](https://imgur.com/a/9M7xmqd) , $\mu\rho-BALD$ preferes points with low, but sufficient, overlap.

---

> > ### Comment · Reviewer_EmX1 · 2021-08-25
> > **Response to rebuttal**
> >
> > I thank the authors for providing a detailed response to my review.  I too have no further questions, and continue to recommend acceptance.

---

### Official Review · Reviewer_Tcca · 2021-07-16

**Rating:** 7
**Confidence:** 3

**Summary:**

The paper proposes several acquisition functions for performing active learning in a causal setting. The paper explores the limitations of more naive acquisition functions and motivates its own proposals. The performance of the proposed acquisition functions is empirically compared to that of more naive acquisition functions.

**Limitations And Societal Impact:**

Yes.

**Main Review:**

**Originality**: The authors do acknowledge other similar methods; however, I think they fairly outline the differences between them and why they matter. Furthermore, in comparison to other acquisition functions, I think a good case is made for why the proposed functions are far superior. Overall, not wildly original, but original in ways that I was convinced do matter.

**Quality**: I found the work to be technically sound, with careful consideration of the problems that can arise with naive acquisition functions and a logical progression towards more intelligently constructed ones. I found the experimental section convincing. I would have appreciated some comment on the limitations of the method / avenues for future work.

**Clarity**: I found the work to be clearly written and organized. Figures 2 and 3 were very effective.

**Significance**: I believe the work is significant. Active learning is an important field of research, with undeniable potential for social impact. The proposed acquisition functions greatly outperform naive ones and the overall approach does differentiate itself in important ways from competing methods. The proposed acquisition functions can also be used in tandem with other BALD variants that may be developed in the future, furthering their applicability.

**Time Spent Reviewing:**

3

---

> ### Author Response · Authors · 2021-08-10
> **Response to Reviewer Tcca**
>
> We thank the reviewer for their valuable feedback. We address their comments on limitations and future work in the following.
>
> **Limitations**
>
> Reviewer 8K1A brings up some excellent points about unobserved confounding. We believe the unconfoundedness assumption is a significant limitation that we would be happy to discuss further. A summary of our response to Reviewer 8K1A follows for your convenience:
>
> Unobserved confounding can result in biased sampling of training data with respect to the hidden confounding variable. This can result in both performance inequality between groups and a biased estimate of the unconfounded CATE function. Further, the models’ uncertainty estimates are not informative of when this may occur. We will include a discussion on how hidden confounding can lead to biased representation and performance in the space of unobserved variables as well as a biased estimate of the true CATE function.
>
> Another limitation we will highlight is that we are limited by the overlap present in the pool dataset, given that we are not yet considering the case where active treatment assignment is possible.
>
> **Future Work**
>
> Important avenues for future work could be relaxing the unconfoundedness assumption, or, given that it is untestable without further assumptions, incorporating beliefs about hidden confounding into the acquisition function. Furthermore, we think it would be interesting to revisit the case where treatment assignment is considered in addition to uncovering the outcome. Finally, on the active learning side, exploring more rigorous batch acquisition methods could yield improvements over the current stochastic sampling estimation we use.

---

> > ### Comment · Reviewer_Tcca · 2021-08-24
> > **Response to Rebuttal**
> >
> > I thank the authors for their response and for introducing the above comments into the discussion. I will keep my accept recommendation.

---

### Author Response · Authors · 2021-08-10
**To All Reviewers**

We thank all reviewers for the detailed feedback, and are excited about the new conference guidelines encouraging reviewers to have an active discussion with the authors during the discussion period. We believe this will help resolve any remaining misunderstandings in the reviews or our response.

We are pleased that the reviewers agree that we explore an “interesting and important” [EmX1] problem in a “non-trivial, relevant, and understudied setting.” [8K1A] Furthermore, they find our methods are “well motivated,” [wzW6] and “original in ways that .. do matter.” [Tcca, 8K1A] Finally, they find our work is well-written  and clear in its exposition.” [EmX1]

Please see below where we ask specific questions to clarify ambiguities in the reviews, and address individual reviewers' comments.

---

### Decision · Program_Chairs · 2021-09-27

**Decision:**

Accept (Poster)

**Comment:**

The paper aims to provide a sample efficient strategy to train deep learning models that can effectively predict individual-level treatment effects from high dimensional observation data. A unique challenge in estimating personalized treatment effect is the necessity to choose data samples from regions with support from both treated and control populations so that the treatment effect can be properly identified. As a result, most existing data acquisition functions that primarily focus on uncertainty become less effective.  The proposed model, referred to as Causal-BALD, addresses this challenge by integrating Bayesian active learning and causal inference in novel ways that cover the uncertainty and overlapping criteria simultaneously. Effectiveness of the proposed acquisition functions is clearly demonstrated using both synthetic and semi-synthetic datasets.

As acknowledged by all the reviewers, the proposed acquisition functions are clearly motivated, well explained, sufficiently novel, and further supported by convincing evaluation results. It has the potential to impact both the scientific community and multiple application domains.
The authors adequately addressed most of the questions raised by reviewers in their original reviews during the discussion period. In particular, since multiple acquisition functions are proposed and some show better performance than others on certain datasets, it is important for the authors to offer some deeper insight to better understand the behaviors of different acquisition functions. It is also interesting to show some sampled data instances and connect them with the corresponding acquisition function. The authors promised to move some results to the appendix and add necessary discussions to better highlight the key properties of the acquisition functions and their potential limitations.